# A Machine Learning Approach for Chronic Heart Failure Diagnosis

**DOI:** 10.3390/diagnostics11101863

**Published:** 2021-10-10

**Authors:** Dafni K. Plati, Evanthia E. Tripoliti, Aris Bechlioulis, Aidonis Rammos, Iliada Dimou, Lampros Lakkas, Chris Watson, Ken McDonald, Mark Ledwidge, Rebabonye Pharithi, Joe Gallagher, Lampros K. Michalis, Yorgos Goletsis, Katerina K. Naka, Dimitrios I. Fotiadis

**Affiliations:** 1Department of Biomedical Research, Institute of Molecular Biology and Biotechnology, FORTH, 45110 Ioannina, Greece; daphni.plati@gmail.com (D.K.P.); etripoliti@gmail.com (E.E.T.); goletsis@uoi.gr (Y.G.); 22nd Department of Cardiology, Faculty of Medicine, School of Health Sciences, University of Ioannina, 45110 Ioannina, Greece; md02798@yahoo.gr (A.B.); aidrammos@yahoo.gr (A.R.); iliadadimou@gmail.com (I.D.); ftpcavalier52@gmail.com (L.L.); lamprosmihalis@gmail.com (L.K.M.); drkknaka@gmail.com (K.K.N.); 3Wellcome-Wolfson Institute for Experimental Medicine, Queen’s University, Belfast BT9 7BL, UK; chris.watson@qub.ac.uk; 4University College Dublin, National University of Ireland, Belfield, D04 Dublin, Ireland; kenneth.mcdonald@ucd.ie (K.M.); mark.ledwidge@ucd.ie (M.L.); rpharithi@gmail.com (R.P.); jgallagher@ucd.ie (J.G.); 5Department of Economics, University of Ioannina, 45110 Ioannina, Greece

**Keywords:** heart failure, machine learning

## Abstract

The aim of this study was to address chronic heart failure (HF) diagnosis with the application of machine learning (ML) approaches. In the present study, we simulated the procedure that is followed in clinical practice, as the models we built are based on various combinations of feature categories, e.g., clinical features, echocardiogram, and laboratory findings. We also investigated the incremental value of each feature type. The total number of subjects utilized was 422. An ML approach is proposed, comprising of feature selection, handling class imbalance, and classification steps. The results for HF diagnosis were quite satisfactory with a high accuracy (91.23%), sensitivity (93.83%), and specificity (89.62%) when features from all categories were utilized. The results remained quite high, even in cases where single feature types were employed.

## 1. Introduction

HF is a clinical syndrome of various etiologies, in which the heart cannot pump enough blood to satisfy the metabolic needs of the body [1]. Patients with HF have to deal with changes that severely affect their quality of life. HF is one of the major causes of mortality [2] and the most common cause of hospital admissions in people over 65 years of age [3]. Projections show that the prevalence of HF will increase by 46% from 2012 to 2030, resulting in >8 million people ≥18 years of age with HF [4]. Based on the left ventricular ejection fraction (EF), HF patients are classified as HF with reduced EF (HFrEF; EF < 40%), mid-range EF (HFmrEF; EF 40–49%), and preserved EF (HFpEF; EF ≥ 50%) [5].

The diagnosis of HF can be challenging, especially in the early stages and in patients with HFpEF. Symptoms and signs may be particularly difficult to identify and interpret in obese individuals, in the elderly, and in patients with chronic lung disease [3]. Along with standard laboratory investigations, echocardiogram, electrocardiogram (ECG), and natriuretic peptides are probably the most useful tests for diagnosis in patients with suspected HF. For the management of HF, the European Society of Cardiology (ESC) Guidelines [5] propose a range of approaches, including medical management with renin-angiotensin–aldosterone system therapies, beta blockers, mineralocorticoid receptor antagonists, sodium–glucose co-transporter 2 inhibitors (for HFrEF) and diuretic therapies, careful management of cardiovascular and other co-morbidities, biomarker monitoring with serial natriuretic peptide measurements, remote monitoring (using an implanted device when indicated), structured telephone support, and multidisciplinary care. For most patients, the standard management of HF involves office-based follow-up 2–12 times a year, while patients are advised to monitor their weight, blood pressure, pulse, diet, and symptoms on a daily basis.

ML offers the potential to improve healthcare efficiency in numerous ways. Prognostic models may empower healthcare experts to select better treatment options for their patients. Additionally, diagnostic models can be used in screening, in risk stratification, and in recommending appropriate tests. This decreases the burden on clinicians, saves resources, and reduces costs. Due to the increased incidence and the large financial costs associated with the management of HF, the diagnosis and treatment of the disease remain extremely important issues.

Several studies have been conducted to build a model that can diagnose HF based on various ML algorithms. Ali et al. [6], Javeed et al. [7], Samuel et al. [8], Mohan et al. [9], and Potter et al. [10] utilized the Cleveland Heart Disease Database that consists of demographics, symptoms, clinical and laboratory values, and electrocardiographic features. Choi et al. [11] detected HF on a multivariate dataset consisting of demographics, habits, clinical and laboratory values, the International Classification of Disease version 9 (ICD-9) codes, information in Current Procedural Terminology (CPT) codes, and medication features. Son et al. [12] tested a rough set (RS)-based model on demographic characteristics and clinical laboratory values. Reddy et al. [13] detected HFpEF by analyzing medications, demographics, comorbidities, and echocardiographic and ECG features. Masetic et al. [14], Acharya et al. [15], and Ning et al. [16] analyzed ECG signals to detect HF. Lal et al. [17], Wang et al. [18], Chen et al. [19], and Gladence et al. [20] utilized Heart Rate Variability (HRV) measures to diagnose congestive HF. Zheng et al. [21] and Gjoreski et al. [22] suggested a system for chronic HF diagnosis based on the analysis of heart sound characteristics. In Table 1, all studies mentioned in the literature review are presented in detail in order to discriminate between different approaches, methods, and datasets.

All previous works focus on classification between HF and non-HF, using various methods, datasets, and features. Such a classification, although very useful for an automated diagnosis system, provides limited support to an experienced clinician that lacks the ability to perform laboratory tests and echocardiogram due to various logistic reasons [23,24]. In the present study, we propose a methodology to diagnose HF; its main characteristic is that the models are based on various combinations of features, using the clinical approach followed by clinicians, based on current guidelines [5]. In order to examine how each feature type contributes to the diagnosis, initially, our models were built by utilizing only clinical features, i.e., features that can be collected by all clinicians without performing laboratory tests or echocardiogram, such as the patient’s medical history, results from the physical examination, symptoms, comorbidities, demographic information, etc. Subsequently, various combinations of features are assessed by adding additional types of features: Clinical features and natriuretic peptides, clinical and echocardiographic features, echocardiographic features exclusively, and finally all features combined. In this way, we examined how—through the application of machine learning—the different types of features proposed in the guidelines can be applied for HF diagnosis. The results can also be of special value in cases where some of the features cannot be easily obtained.

## 2. Materials and Methods

### 2.1. The Dataset

Data were provided by the University College Dublin (UCD), Ireland (410 subjects), and the 2nd Department of Cardiology of the University Hospital of Ioannina (77 subjects). The total number of subjects was 487 (260 without HF, 180 with chronic HF, and 47 with acute HF). As is common in ML studies, the maximum available data were used (available patient data from patients having accepted informed consent). In order to train and test ML algorithms, both kinds of subjects are needed, i.e., HF and controls. This allows algorithms to be trained for both cases and to be able to correctly classify any new case. The same applies for testing; testing needs to be done for both cases (HF/non-HF). In our case, we had 227 HF patients and 260 controls. Patients were diagnosed with HF by clinical experts. This diagnosis was based on the patients’ physical examinations, along with standard laboratory investigations, echocardiograms, ECGs, and the measurement of natriuretic peptides. The features recorded for each patient were grouped into the following categories: General demographic data, classical cardiovascular risk factors, personal history of cardiovascular disease, other diseases, lifestyle/habits, medications, symptoms, physical examination, laboratory findings, and echocardiographic features (Table 2).

Certain features were removed due to overlapping information, i.e., interventricular septal thickness and posterior wall thickness at the end diastole had similar information to the left ventricular mass index values. NYHA classification and dyspnea symptoms were recorded in all patients, but were not included in the HF classification analysis. Rhythm device information was also excluded from the final analysis, since the target population who may benefit from the application of these models will not have implanted defibrillators or resynchronization therapy (these therapies are applied to already diagnosed symptomatic NYHA II–IV HF patients). Finally, the classification of HF phenotype (HFpEF, HFrEF, or HFmrEF) did not fit in our analysis, since the models were to be used for HF diagnosis in previously undiagnosed patients. Medications were not included in our final dataset, since they can be considered either as a determinant or as an indicator of a patient’s condition. The resulting dataset is depicted in Table 3.

### 2.2. The Proposed Methodology

The proposed methodology consists of three basic stages: Preprocessing, feature selection, and classification. The preprocessing pipeline includes the removal of features with ≥50% missing values. Discrete features with unbalanced distribution of values are also removed and outliers and typos (e.g., 4,5 is recorded instead of 4.5) per feature are detected and corrected. Furthermore, the class imbalance problem is handled by applying the undersampling method to the dataset, which is quite common for dealing with this issue [25]. This method produces a random subsample with a given spread between class frequencies. The maximum “spread” between the rarest and most common class is specified. In this study, a random undersample of the majority class is followed, so that all classes have the same number of instances. The 19 features retained after removing features with ≥50% missing values and discrete features with unbalanced distribution of values and utilized for the diagnosis of HF are presented in Table 4. For better understanding of the individual contribution of each feature, we calculated the information gain [26] metric as presented in Table A2 of Appendix B.

At the second stage, feature selection is applied to all features from all categories and the subset of features retained is used for the classification process. In this study, feature selection methods are employed to assess the predictive ability of feature subsets and the degree of redundancy among them, preferring sets of features that are highly correlated with the class but with low intercorrelation [27].

Finally, at the classification stage, different classifiers (i.e., decision tree, RF, rotation forest (ROT), naive Bayes (NB), K-NN, SVM, logistic model tree (LMT), and Bayes network (BN)) are applied to the reduced feature subset. Then, 10-fold cross-validation is applied for the evaluation of the classifiers, which is a statistical method for evaluating and comparing learning algorithms by dividing data into two segments: One used to learn or train a model and the other used to validate the model [28]. The proposed methodology is presented in Figure 1. The results are expressed in terms of accuracy, sensitivity, and specificity.

The aforementioned methodology was applied for the following combination of features (Figure 2).

## 3. Results

The mean age of the subjects in the dataset was 69 years and the median age was 71 years. Regarding HF patients, the mean age was 72 and the median was 74 years, while for subjects without HF, the mean age was 67 and median was 68 years. The dataset consisted of 260 male and 224 female subjects. The dataset with documented HF consisted of 73 females and 154 males, while the non-HF dataset consisted of 151 female and 106 male subjects (for three subjects, gender information was unavailable). Furthermore, regarding the subjects with HF, 68 were classified as HFpEF, while 60 as HFrEF and 88 as HFmrEF (for 11 subjects, the classification according to ejection fraction was missing, since the ejection fraction was unavailable).

For the HF diagnosis, all subjects with acute HF and subjects with NYHA classification III–IV were removed, as diagnosis of HF in patients with severe symptoms is not challenging. Thus, the dataset for HF diagnosis consisted of 422 subjects (260 without HF and 162 with chronic HF).

The results for the HF diagnosis problem with feature selection are depicted in Table 5.

Our approach achieved the highest results in terms of accuracy using mostly the LMT and ROT classifiers with different combinations of features included. The accuracy values ranged from 84.12% in models using only clinical features to 91.23% in models using all features combined.

The optimal features that were retained after the feature selection procedure in various models with different feature combinations are presented in Table 6. All retained features showed statistically significant (*p* < 0.01) correlations (Figure A1 and Table A3 and Table A4) with the diagnosis of HF.

## 4. Discussion

Data-driven approaches for the optimization of population health management are continuously growing and may prove valuable in modern healthcare models, especially for highly prevalent and costly diseases such as HF. The present study made another considerable contribution toward the development of such an approach for the diagnosis of HF in symptomatic patients with risk factors based on simple clinical data, as well as natriuretic peptides and echocardiographic indices (suggested by ESC guidelines) with the use of various machine learning techniques. The results for HF diagnosis were quite high in terms of accuracy (91.23%), as well as in terms of sensitivity (93.83%) and specificity (89.62%), confirming the classification power of ML approaches. Furthermore, our model achieved high accuracy, even when only the clinical features were used for classification (84.12%), which can prove to be of great value for an initial screening in settings where laboratory tests are not available. We also noticed that the addition of BNP to clinical features increased the accuracy of HF diagnosis (88.15%), as expected based on the well-established value of natriuretic peptides in the diagnosis of HF [5]. Furthermore, the combination of clinical and echocardiographic features for the classification of HF diagnosis also resulted in increased accuracy compared to clinical features alone (accuracy 90.76%). This finding re-emphasizes the diagnostic value of echocardiography that, even without natriuretic peptides, can establish the diagnosis of HF in the majority of patients with suspected non-acute HF [5]. A small difference in terms of accuracy between models using BNP or echocardiographic parameters was observed; whether these differences are of clinical importance is not known.

Furthermore, in every classification model, our method finally utilized a smaller feature subset. This may indicate that a small number of clinical, biochemical, and echocardiographic parameters are needed for reaching a diagnosis of HF, which corresponds to less time and cost. From a clinical point of view, the current study suggests that the identification of a few classic risk factors (e.g., hypertension) and common cardiovascular diseases (e.g., coronary artery disease and atrial fibrillation) are most important in the diagnosis of HF. Regarding the laboratory tests taken into account, BNP (and its *N*-terminal counterpart NTproBNP) is the biomarker most widely used. It is secreted mainly by the ventricular heart muscle and causes natriuresis, diuresis, and smooth muscle relaxation. It is increased in HF of all etiologies [29], both in acute and chronic settings. We might have also used troponin T (TnT), a polypeptide forming part of the contractile apparatus of the striated muscle, which is the best laboratory parameter in the early diagnosis of acute myocardial infarction and also has a predictive role in various diseases of the cardiovascular system, such as HF and/or in hemodynamic instability [30,31]. The latest guidelines by the ESC recommend the use of troponin for the exclusion of an acute coronary syndrome and suggest it for estimating the risk of myocardial damage in other situations such as hereditary muscle diseases (dilated, hypertrophic, and arrhytmogenic cardiomyopathy), cardio-toxic medications used in oncology, myocarditis, and atrial diseases [5]. However, in chronic HF, TnT tends to be less marked and it is used as incremental information to natriuretic peptides [32]. For simplicity reasons we included only the biomarker that was incorporated in the diagnostic algorithm for HF proposed by the ESC [5].

Besides natriuretic peptides, the use of only a few structural echocardiographic indices (such as LV mass, left atrial size, and LV ejection fraction) may be also valuable for HF diagnosis. In this way, ML may assist in the isolation of high-risk features and thus provide an appropriate phenotyping that may improve the detection accuracy of HF. These methods may also lead to greater insights into the pathophysiological pathways underlying the development of HF and the design of future clinical studies that will validate the clinical importance of our findings.

Relevant approaches in the literature provide a method for detecting HF based on several feature types with an accuracy ranging from 72.44% to 99.22%. Our method cannot be directly compared with those utilizing ECG signals [14,15,16], HRV measures [17,18,19], or heart sound characteristics [21,22], but with those that utilize the Cleveland Heart Disease Database [6,7,8,9,10], which is a dataset that resembles ours, and studies that use a multivariate dataset [11,12,13]. It should be noted that the present study utilized a larger dataset (422 instances) compared to the studies that utilized the Cleveland Heart Disease Database. Moreover, in our approach, the medications were not finally considered in the feature set. If we included medications, the obtained results would further increase (the ROT classifier achieved 93.36% accuracy, 95.70% sensitivity, and 91.90% specificity). Still, as the addition of medications might introduce a kind of bias, this approach was not selected. We also tested whether CAD and Arr-Afib could be omitted, as they are not necessarily known or easy to determine during a consultation. It seems that these two features can slightly contribute to the performance of our classifier. In more detail, by omitting these two features the evaluation metrics (namely, the accuracy, sensitivity, and specificity) changed from 91.23%, 93.83%, and 89.62% to 90.28%, 94.00%, and 85.00% (Table A1). On the other side, this is an indication that our approach works adequately, even without these hard-to-obtain features.

Moreover, we excluded several features (NYHA class, device, dyspnea, and HF phenotype) and subjects with acute HF and NYHA classes III–IV as they could be indicative of HF presence. In our study, feature selection was applied, concluding to a smaller feature set where all retained features were significantly correlated with the class (Table A2 and Table A3); even with a smaller feature, set the achieved results were high. This study provides an automated diagnostic tool with high accuracy for detecting the presence of HF, even in cases when limited tests (echocardiogram and laboratory tests) are offered. Additionally, it can be valuable in cases when multiple co-morbidities occur and can offer the clinical expert a further aid in the diagnosis of HF.

Limitations: Although the current study was performed with one of the largest datasets compared to the literature, the incorporation of the proposed approach in a Clinical Decision Support System used in actual clinical practice requires extensive testing and validation with a larger and more diverse dataset.

## 5. Conclusions

In the present study, we developed a method approach able to diagnose the presence of HF based on ML techniques. This study is quite innovative, because we simulated the clinical procedure and investigated the impact of different feature types on the classification accuracy. The results for the HF diagnosis, when all available feature types were utilized for classification, were high in terms of accuracy (91.23%), sensitivity (93.83%), and specificity (89.62%). Efficiency is supported as a limited feature set is selected through feature selection, minimizing the need for diagnostic tests. Moreover, even without the whole feature set, our approach provides quite high results; the results remain high even when only clinical features are used. This provides opportunity to clinicians that do not have the opportunity to perform laboratory tests or echocardiograms to diagnose HF quite accurately without necessarily needing the input of additional tests.

## Figures and Tables

**Figure 1 diagnostics-11-01863-f001:**
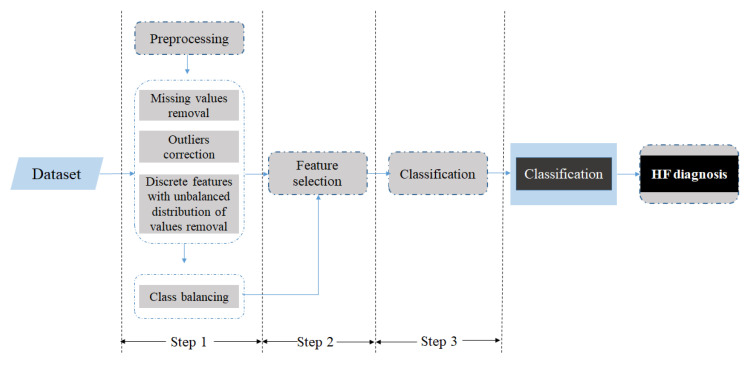
Methodology.

**Figure 2 diagnostics-11-01863-f002:**
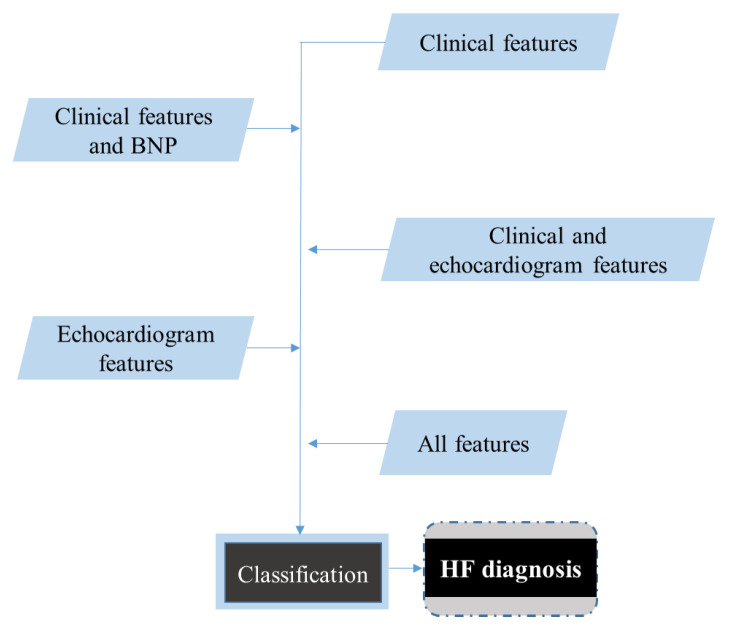
Classification for various sets of features.

**Table 1 diagnostics-11-01863-t001:** State of the art in machine learning for HF diagnosis.

Study	Target	Method	Features	Dataset	Measures
Zheng et al. [21] (2015)	Chronic HF diagnosis Healthy vs. chronic HF	Least square-Stacked Support Vector Machine (SVM) model	Cardiac reserve and heart sound characteristics	152 subjects 88 controls 64 chronic with HF	Acc 95.39% Sens 96.59% Spec 93.75%
Masetic et al. [14] (2016)	Congestive HF diagnosis Healthy vs. congestive HF	Decision tree, K-Nearest Neighbors (K-NN), SVM, Neural Network (NN), and Random Forest (RF)	ECG signals	31 subjects 18 with congestive HF 13 controls	RF acc 100%
Choi et al. [11] (2017)	HF diagnosis Healthy vs. HF	Recurrent Neural Network (RNN) models, Logistic Regression (LR), SVM, Multilayer Perceptron (MLP), K-NN	Demographics, habits, clinical and laboratory values, ICD-9 codes, CPT codes, and medications	3884 with HF 28.903 controls	RNN model AUC 77.70%
Chen et al. [19] (2017)	Congestive HF diagnosis Healthy vs. congestive HF	Deep Neural Network (DNN)	HRV measures based on the RR interval	116 subjects 44 with congestive HF 72 controls	Acc 72.44% Sens 50.39% Spec 84.93%
Samuel et al. [8] (2017)	HF diagnosis Healthy vs. HF	Hybrid decision support method based on artificial neural networks and fuzzy analytic hierarchy process (Fuzzy_AHP) techniques	Demographics, symptoms, clinical and laboratory values, and electrocardiographic results	Cleveland heart disease database 297 subjects 137 with HF 160 controls	Acc 91.10%
Reddy et al. [13] (2018)	HFpEF identification	LR	Medications, demographics, comorbidities, and echocardiographic and ECG features	414 subjects 267 with HFpEF 147 controls	AUC 88.60%
Wang et al. [18] (2019)	Congestive HF diagnosis Healthy vs. congestive HF	Combination of the Long Short-Term Memory (LSTM) network and convolution net architecture	HRV measures based on the RR interval	156 subjects 44 with congestive HF 112 controls	Acc 99.22%
Acharya et al. [15] (2019)	Congestive HF diagnosis Healthy vs. congestive HF	Convolutional neural network (CNN)	ECG signals	73 subjects 15 with congestive HF 58 controls	Acc 98.97% Spec 99.01% Sens 98.87%
Ali et al. [6] (2019)	HF diagnosis Healthy vs. HF	SVM	Demographics, symptoms, clinical and laboratory values, and electrocardiographic results	Cleveland heart disease database 297 subjects 137 with HF 160 controls	Acc 92.22% Sens 100.00% Spec 82.92%
Javeed et al. [7] (2019)	HF diagnosis Healthy vs. HF	Random Search Algorithm (RSA) for feature selection and RF for classification	Demographics, symptoms, clinical and laboratory values, and electrocardiographic results	Cleveland heart disease database 297 subjects 137 with HF 160 controls	Acc 93.33%
Mohan et al. [9] (2019)	HF diagnosis Healthy vs. HF	Hybrid RF	Demographics, symptoms, clinical and laboratory values, and electrocardiographic results	Cleveland heart disease database 297 subjects 137 with HF 160 controls	Acc 88.40% Sens 92.80% Spec 82.60%
Lal et al. [17] (2020)	Congestive HF diagnosis Healthy vs. congestive HF	SVM Gaussian, K-NN, decision tree, SVM RBF, and SVM polynomial	HRV measures	116 subjects 44 with congestive HF 72 controls	SVM Gaussian Acc 88.79% Sens 93.06% Spec 81.82% AUC 95.00%
Gjoreski et al. [22] (2020)	Chronic HF diagnosis Healthy vs. chronic HF	Combination of classic ML and end-to-end Deep Learning (DL)	Heart sound characteristics	947 subjects	Acc 92.90% Sens 82.30% Spec 96.20%
Potter et al. [10] (2020)	Stage B HF detection	RF	Demographics, symptoms, clinical and laboratory values, and electrocardiographic results	Cleveland Heart Disease Database 254 subjects as train set (135 with HF, 119 controls) 65 subjects as test set (27 with HF, 38 controls)	AUC 76.00% Sens 93.00% Spec 61.00%
Ning et al. [16] (2020)	Congestive HF diagnosis Healthy vs. congestive HF	Hybrid DL algorithm that is composed of a CNN and a recursive NN	ECG signals	33 subjects 15 chronic HF subjects 18 controls	Acc 99.93% Sens 99.85% Spec 100%

**Table 2 diagnostics-11-01863-t002:** Initial dataset provided.

Category	Description
General demographic data	Age and gender
Classical cardiovascular risk factors	Hypertension and diabetes mellitus
Personal history of cardiovascular disease	Device, myocardial infarction (MI), coronary artery disease (CAD), angina, peripheral vascular disease, any arrhythmia (Arr), paroxysmal atrial fibrillation (Afib), and stroke
Other diseases	Arthritis, chronic obstructive pulmonary disease, cancer, asthma, gout
Lifestyle/habits	smoking, and physical activity
Medications	Mineralocorticoid receptor antagonists (MRAs), diuretics (loop or thiazide diuretic), calcium channel blocker (CCB), statin, antiplatelet, renin angiotensin aldosterone system (RAAS), beta blocker (BB), oral anticoagulant (OAC), other lipid-lowering drugs (LipD), alpha blocker, digoxin, insulin, warfarin, nitrate, diabetes drugs, and ivabradine
Symptoms	Dyspnea, orthopnea, NYHA classes I–IV, and paroxysmal nocturnal dyspnea
Physical examination	Weight, height, body mass index (BMI), murmurs, systolic blood pressure (SBP), diastolic blood pressure (DBP), heart rate (HR), pulse, crackles, oedemas, JVP distension, and body surface area
Laboratory findings	BNP, Na, K, Ca, Cl, urea, creatinine, eGFR, full blood count including WBC, full blood count including Hb, platelet count, total cholesterol, HDL, LDL, triglycerides, and glucose (non-fasting)
Echocardiographic parameters	Interventricular septal thickness at end-diastole (IVS), posterior wall thickness at end diastole (PW), left ventricular internal dimension in diastole (LVIDd), LV mass, left ventricular mass index (LVMI), left atrial volume (average 4ch and 2ch) (LAVI), left atrial (LA) dimension (mm), peak E-value, peak A-value, early filling (E wave)/late diastolic filling (A wave) ratio (E/A), mitral annular velocity (E’), early filling (E wave)/mitral annular velocity(E/E’), Ε deceleration time, ejection fraction (EF), diastolic biventricular inner dimension, estimation of any valvular disease, right ventricular systolic pressure, and pulmonary artery systolic pressure; classification of HF phenotype into: HFrEF, HFmrEF, and HFpEF

**Table 3 diagnostics-11-01863-t003:** Dataset after deleting features.

Category	Description
General demographic data	Age and gender
Classical cardiovascular risk factors	Hypertension and diabetes mellitus
Personal history of cardiovascular disease	MI, CAD, angina, peripheral vascular disease, Arr, Afib, and stroke
Other diseases	Arthritis, chronic obstructive pulmonary disease, cancer, asthma, and gout
Lifestyle/habits	Smoking and physical activity
Symptoms	Orthopnea, Paroxysmal Nocturnal Dyspnea
Physical examination	Weight, height, BMI, murmurs, SBP, DBP, HR, pulse, crackles, edemas, JVP distension, and body surface area
Laboratory findings	BNP, Na, K, Ca, Cl, urea, creatinine, eGFR, full blood count including WBC, full blood count including Hb, platelet count, total cholesterol, HDL, LDL, triglycerides, and glucose (non-fasting)
Echocardiographic parameters	LVIDd, LV mass, LVMI, LAVI, left atrial dimension (mm), peak E-value, peak A-value, EA, mitral annular velocity, Ee, Ε deceleration time, EF, diastolic biventricular inner dimension, estimation of any valvular disease, right ventricular systolic pressure, and pulmonary artery systolic pressure

**Table 4 diagnostics-11-01863-t004:** Features for HF diagnosis.

Category	Description
General demographic data	Age and gender
Classical cardiovascular risk factors	Hypertension
Personal history of cardiovascular disease	MI, CAD, and any arrhythmia (Arr) or paroxysmal atrial fibrillation (Afib) combined as Arr-Afib
Physical examination	BMI, SBP, DBP, and HR
Laboratory findings	BNP
Echocardiographic parameters	LVIDd, LVMI, LAVI, EA, E deceleration time, Ee, EF, and peak E-value

**Table 5 diagnostics-11-01863-t005:** HF diagnosis classification results.

Features Type	Classifier	Accuracy %	Sensitivity %	Specificity %
Clinical features	LMT	84.12	82.10	85.38
Clinical features and BNP	LMT	88.15	85.80	89.62
Clinical and echocardiographic features	ROT	90.76	93.21	89.23
Echocardiographic features	ROT	87.91	90.74	86.15
All features	ROT	91.23	93.83	89.62

**Table 6 diagnostics-11-01863-t006:** Retained features for HF diagnosis, all possible features set.

Feature Set	Retained Features
**Clinical features**	Hypertension, Arr-Afib, CAD, and SBP
**Clinical features and BNP**	Hypertension, Arr-Afib, CAD, SBP, and BNP
**Clinical and echocardiogram features**	Hypertension, Arr-Afib, CAD, SBP, EF, LAVI, LVMI, E/E’, and E deceleration time
**Echocardiogram features**	EF, LAVI, LVMI, and E deceleration time
**All features**	Hypertension, Arr-Afib, LAVI, LVMI, CAD, BNP, SBP, and EF

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
