# Peer review of "A Machine Learning Approach for Chronic Heart Failure Diagnosis"

_diagnostics, 2021, doi:10.3390/diagnostics11101863_

Round 1

Reviewer 1 Report

The authors have addressed the issues raised previously and hence the manuscript can be considered for publication. 

Author Response

Thank you for your comments.

Reviewer 2 Report

Thank you for the opportunity to review your manuscript entitled

A machine learning approach for Chronic Heart Failure diagnosis’’

Troponin T (TnT) is a polypeptide forming part of the contractile apparatus of the striated muscle. The TnT function in all types of striated muscles is the same. As one of the three proteins included in the troponin complex, it performs key functions in the process of muscle contraction. A very important aspect from a diagnostic point of view is the fact that the sequence of troponins of cardiac origin differs from the sequence of skeletal muscle troponins. Considering the fact that the cytoplasm of cardiomyocytes contains a small amount of free troponins TnT and Troponin (Tnl), even small damage to the cell membrane (e.g., in the initial phase of myocardial ischemia) causes their release and the ability to be detected in the blood sample. Thanks to this, after obtaining specific monoclonal antibodies, it became possible to use them in the diagnosis of ischemic heart disease. Currently, it is believed that cardiac TnT is the best laboratory parameter in the early diagnosis of acute myocardial infarction. In the available literature, numerous articles describe hs-TnT as a predictive biomarker in various diseases of the cardiovascular system such as heart failure and hemodynamic instability (1,2).

Abstract, title and references.

The aim of the study is clear. The title is informative and relevant. The references are relevant, recent, and referenced correctly. Please complete your references with the following article:  1. DOI: 10.2217/bmm-2018-0186 2. DOI: 10.1097/SHK.0000000000001360 

Introduction

It is clear what is already known about this topic. The research question is clearly outlined.

Methods

The process of subject selection is clear. The variables are defined and measured appropriately. The study methods are valid and reliable. There is enough detail in order to replicate the study.

Discussion and Results

The results are discussed from multiple angles and placed into context without being overinterpreted. The conclusions answer the aims of the study. The conclusions supported by references and results. The limitations of the study are opportunities to inform future research.
Overall. The study design was appropriate to answer the aim. The manuscript is well written and a stimulus for the readership.

Minor revisions:

  1. Did the study evaluate the level of Troponin T?

  1. Did the study take into account the influence of Troponin T on the occurrence of the primary endpoint?
  2. Please add the following reference:

  1. DOI: 10.2217/bmm-2018-0186
  2. DOI: 10.1097/SHK.0000000000001360

Author Response

Thank you for your comments.

  1. Did the study evaluate the level of Troponin T?

  1. Did the study take into account the influence of Troponin T on the occurrence of the primary endpoint?

The following has been added in Discussion section, page 11.

Regarding the laboratory tests taken into account, BNP (and its N-terminal counterpart NTproBNP) is the biomarker most widely used. It is secreted mainly by the ventricular heart muscle and causes natriuresis, diuresis and smooth muscle relaxation. It is increased in HF of all etiologies [Zaphiriou et al. 2005] both in acute and chronic setting. We might have also used Troponin T (TnT), a polypeptide forming part of the contractile apparatus of the striated muscle, which is the best laboratory parameter in the early diagnosis of acute myocardial infarction and has also a predictive role in various diseases of the cardiovascular system such as HF and/or in hemodynamic instability [Duchnowski et al. 2018], [Duchnowski et al. 2020]. The latest guidelines by the European Society of Cardiology recommend the use of troponin for the exclusion of an acute coronary syndrome and suggest it for estimating risk of myocardial damage in other situations like hereditary muscle diseases (dilated, hypertrophic and arrhytmogenic cardiomyopathy), cardio-toxic medications used in oncology, myocarditis and atrial diseases [McDonagh et al. 2021]. However, in chronic HF, TnT tends to be less marked and it is used as incremental information to natriuretic peptides [Eggers et al. 2017]. For simplicity reasons we included only the biomarker that has been incorporated in the diagnostic algorithm for HF proposed by the ESC [McDonagh et al. 2021]. (See Discussion section, page 11.)

  1. Please add the following reference: DOI: 10.2217/bmm-2018-0186, DOI: 10.1097/SHK.0000000000001360

The proposed references have been added.

 Also we have added in the manuscript the latest ESC guidelines (August 2021 McDonagh doi:10.1093/eurheartj/ehab368).

This manuscript is a resubmission of an earlier submission. The following is a list of the peer review reports and author responses from that submission.

Round 1

Reviewer 1 Report

Manuscript Number: diagnostics-1231537

The manuscript by Plati et. al. and coauthors entitled, “A machine learning approach for Chronic Heart Failure diagnosis” aimed at addressing the chronic heart failure (HF) diagnosis with the application of machine learning approaches.  The models build by the authors are based on various combinations of features categories i.e. clinical features, echocardiogram and laboratory findings.  The results of the study showed high accuracy (91.23%), sensitivity (93.83%) and specificity (89.62%) for HF diagnosis.

This study is mostly exploratory in nature and the study design is good. Few minor issues in the manuscript need to be addressed before considering it for publication.

  1. Introduction needs to be revised as it is sounding more of a review of literature. Most importantly, the rationale of the study is unclear.

  1. Results: male and female patients don’t add up and match total number of patients used in the study. Similarly, total HF patients (both male and female) are not equal to HF subjects classified according to different Ejection Fractions. There is strong discrepancy in subject number.

  1. Table 6 is nowhere mentioned in the manuscript. It is advisable to mention about the utility and the table in the text.

Reviewer 2 Report

The aim of the study submitted by Plati and coll is to address the chronic heart failure diagnosis with the application of machine learning approaches simulating the algorithm followed in clinical practice.

Major issues

The introduction needs substantial reduction; almost halved, I’d say. Most of the information provided is non-essential and not necessary to clarify the premises of the study.

Since the aim of the study is the development of a machine learning approaches simulating the algorithm for the diagnosis of heart failure, why of the 487 patients enrolled, 260 did NOT have heart failure? Was a sample size estimated?

The entire article is built in an anomalous way, not respecting the typical structure of an original article. It is extremely difficult to read and fails to convey a clear message with respect to the advantage of the methodology proposed by the authors. The advice is to lighten it and simplify it considerably. An alternative suggestion could be to turn it into a review with original data added.

Minor concerns

Occasionally in the text you will find the words “(Error! Reference source not found.)”: please delete.

The legend Is missing in the tables;  table 6 is misplaced, it is located between the discussion section and the conclusion section

Reviewer 3 Report

In this study, Plati et al. used machine learning to construct predictive models for heart failure diagnosis based on clinical presentation, BNP levels, and echocardiogram parameters of patients. The strength of this article lies in the predictive accuracy of the models and relative easiness with which parameters can be determined. However, its content is still quite similar to work that has already been published by other researchers. The real value of such a somewhat replicative study would be to test these new algorithms on datasets from already published studies. Maybe this would be interesting for a follow-up study.   

Comments:        

Regarding introduction, please shorten and lead more clearly through data that already exists and what your study adds.

Page 2, Line 51. Based on which criteria were patients diagnosed with heart failure. For example, if diagnosis was based on echocardiogram, it is not surprising that echocardiogram parameters are predictive of heart failure in this sample.

Page 2, Line 83: There are already a lot of studies with predictive models around. The novelty aspect of your study is somewhat questionable. The Cleveland heart disease database seems fairly similar to your approach. What would be really interesting to see is whether your algorithms have a high predictive accuracy in their dataset. But may be such an undertaking might be out of reach. In general, wouldn`t it be more informative to test the predictive accuracy of your model in an unrelated sample? That the model works for the sample it was trained upon is good, but testing the predictive accuracy in an unrelated sample would be more interesting. If your methods already include a training and a test dataset, please mention this in the method section for better understandability.

Page 6, Table 2: Please write out what the abbreviations mean under all tables

Page 6, Table 3: So what your data shows is that if practitioners know the age, gender, BMI, SBP, DBP, HR and whether the patient has hypertension, MI, CAD, or Arr-Afib they could quite accurately predict whether a patient has heart failure. Almost too good to be true. Is there any parameter on the individual contribution of each feature to the predictive power of the model? I presume that a history of MI is more related to heart failure than gender, for example. If available please include in appendix. In that line it would also be interesting to see how good the prediction is when CAD and Arr-Afib are left out of the model as they are not necessarily known or easy to determine during a consultation.

Page 7, Line 209: Please include the actual correlations in a table in the appendix. It would also be interesting to compare the results of your machine learning approach to a simple linear regression model with the clinical features as predictors and HF as dependent variable.  

Please add a limitation section and highlight the limited sample size as major criterion.

Minor points:

Page 2, Line 65: “Error! Bookmark not defined.” Please correct in the whole text.

“Error, Ref. not found” in several lines. Please check.

Please check whole manuscript by a native-speaker to improve written English.

Please re-organize your tables more clearly. For example, table 1 is hard to read as alle text blocks are centered. Figures are blurred. Please increase dpi.